# Parental Feeding Practices and Children’s Eating Behaviours: An Overview of Their Complex Relationship

**DOI:** 10.3390/healthcare11030400

**Published:** 2023-01-31

**Authors:** Alexandra Costa, Andreia Oliveira

**Affiliations:** 1EPIUnit—Instituto de Saúde Pública da Universidade do Porto, 4050-600 Porto, Portugal; 2Laboratório para a Investigação Integrativa e Translacional em Saúde Populacional (ITR), Universidade do Porto, 4050-600 Porto, Portugal; 3Faculdade de Medicina da Universidade do Porto, Alameda Professor Hernâni Monteiro, 4200-319 Porto, Portugal

**Keywords:** feeding behaviours, feeding practices, appetite, child obesity, parents

## Abstract

Several studies have found an association between eating behaviours and weight status and obesity risk in childhood. Children’s eating behaviours arise from a combination of genetic and environmental factors. Parents appear to play a central role in their development as the main responsible for shaping children’s feeding environment and eating experiences. The purpose of this paper is to review the literature on parental influences on eating behaviours across childhood, mainly focusing on parental feeding practices. The associations between parental feeding practices and children’s eating behaviours have been extensively studied. However, most of the findings come from cross-sectional studies, so the possibility of reverse causality cannot be ruled out (i.e., children’s behaviours influencing parents). Most recently, a few longitudinal studies with a cross-lagged design have shown that the relationship between children’s eating behaviours and parental feeding practices seems to be bidirectional, where it is not straightforward whether parental feeding practices are a predictor or a consequence of children’s eating behaviours. Children’s eating behaviours influence parents to adopt certain feeding practices, but these practices also influence children’s behaviours over time. Parental feeding practices may have the potential to shape children’s eating behaviours and should be targeted to promote the development of non-obesogenic traits. However, parent–child interactions are complex and therefore both parent and child characteristics and the family dynamics should be considered.

## 1. Introduction

A growing body of research has focused not only on what children eat, but also how they eat. The “how” is measured by several well-described eating phenotypes, collectively referred to as eating behaviours [1]. Several studies have found an association between these behaviours and weight status and obesity risk in childhood [2]. Children with a more avid appetite (showing behaviours such as food responsiveness, enjoyment of food and emotional overeating, commonly referred to as food approach behaviours) are more likely to have higher body fat and be at risk for excessive weight and obesity [2], whereas those with a smaller appetite (characterized by food avoidant behaviours such as satiety responsiveness, slowness in eating, food fussiness and emotional undereating) are more likely to have lower body fat and be underweight [2]. Food approach behaviours also have a documented association with greater total energy and food intake in children, while food avoidant behaviours have been associated with lower overall food intake, including reduced consumption of fruit and vegetables, and a less varied diet [3,4,5].

Evidence from twin studies shows that eating behaviours have a strong genetic basis [6,7]. Nevertheless, the environment in which children live and grow also plays a crucial role and can interact with genetics. It is believed that individuals respond in divergent ways to the (food) environment they live in, i.e., genetic predispositions to overweight seem to be enhanced in higher-risk home environments [8], which can be defined as a genotype–environment interaction [9]. Several environmental determinants of eating behaviours have been identified, with the early feeding environment and parental feeding practices being the most extensively studied [10]. These are particularly important because they are potentially modifiable factors that can be targeted in parenting interventions [1,10].

Parents are the primary caregivers and play a pivotal role in shaping their children’s feeding environment and eating experiences [11,12,13]. This influence begins even before birth, with exposure in utero, and continues throughout childhood, as parents provide the foods offered and interact with their children during mealtimes. All these aspects have been associated with children’s eating behaviours [14,15,16,17]. However, determining the extent of parental influence on children’s behaviours is complex due to the dynamic nature of the parent–child relationship. Parents may adapt their feeding practices in response to their children’s characteristics, leading to a bidirectional relationship where it is not straightforward whether parental feeding practices are a predictor or a consequence of children’s eating behaviours [18,19,20]. Previous studies have mostly focused on a unidirectional relationship, placing parents as the driving force of influence. However, more recently a bidirectional relationship has been explored, with some studies attempting to clarify the complex interactions between parenting and children’s eating behaviours [14,15,21,22,23].

This paper aims to review the literature on parental influences on children’s eating behaviours. This narrative review will cover the early feeding environment, with a primary focus on parental feeding practices during childhood.

## 2. Methodology

To find relevant studies for this narrative review, we conducted a search on PubMed and employed a snowball identification method. The search strategy consisted of a combination of search terms related to parental feeding practices (e.g., feeding practices, food parenting) and children’s eating behaviours (e.g., eating behaviours, appetitive traits). Our inclusion criteria were observational and review studies, studies performed in children (aged less than 18 years), and studies that aimed to examine the influence of parental practices on children’s eating behaviours. We focused on the most frequently examined eating behaviours and feeding practices. No publication date limits were applied.

## 3. What Are Children’s Eating Behaviours?

Eating behaviours are a set of persistent predispositions and tendencies toward foods that reflect aspects such as hunger, appetite, satiety, and response to food cues [1,24,25]. These behaviours interact with environmental factors and influence food selection and eating choices [1,24]. The Children Eating Behaviour Questionnaire (CEBQ) [26], developed by Professor Jane Wardle, was the first comprehensive psychometric measure to evaluate eating behaviours. It measures eight different dimensions: Desire to drink, enjoyment of food, food responsiveness, emotional overeating, slowness in eating, food fussiness, emotional undereating, and satiety responsiveness. Apart from psychometric measures, eating behaviours can be also assessed using experimental designs and direct observations in laboratory settings (Table 1).

## 4. Parental Influences on Children’s Eating Behaviours

### 4.1. Early Feeding Environment

Parental influence on children’s eating behaviours starts even before birth, with perinatal exposures. The foetus has its first taste experiences through the amniotic fluid, which is favoured by the mother’s diet. Studies have shown that foetal flavour exposure increases acceptance of similarly flavoured foods when re-exposed during infancy and possibly childhood [34]. In milk feeding, the flavour composition of breast milk may vary with maternal diet, whereas infant formula has the same flavour over time [35]. It has been suggested that the sensory experience with human breastmilk influences food acceptance through flavour learning [36]. The type of infant formula may also influence the acceptance of tastes; infants fed with hydrolysed casein formulas, which have pronounced bitter, sour, and savoury tastes, appear to accept new foods more readily later [37,38]. The way each infant is fed, whether directly from the breast or by the bottle, may also be a factor to consider. With bottle feeding, mothers/caregivers have control over the amount of milk offered and the amount ingested by the infant, which could encourage the caregiver to feed an infant regardless of internal hunger and satiety signals [39]. Direct breastfeeding during early infancy has been related to greater appetite regulation in later childhood [40,41]. In a study of Chilean adolescents, shorter breastfeeding duration was associated with poorer satiety response and higher eating in the absence of hunger [42]. However, the associations between breastfeeding and eating behaviours are not consistent. Other studies found no evidence that increased breastfeeding has a lasting and consistent effect on children’s eating behaviours [43,44].

The timing and types of foods introduced during complementary feeding seem also to play a role in the development of eating behaviours [45]. Complementary feeding is a sensitive period for learning food preferences and appetite control, in which infants discover the sensory and nutritional properties of foods [45]. Infants are born with taste predispositions that include rejection of novel foods (food neophobia) and bitter/sour tastes and preference for sweet tastes [46]. When and which foods are introduced during complementary feeding could influence the acceptance of new foods and eating behaviours [47]. Repeated exposure to a variety of foods and textures during the “sensitive period” for introduction to solid foods, between four to five months of age, is thought to promote food acceptance later in life [47,48]. In addition, children who are introduced to a variety of fruits and vegetables early in complementary feeding seem to more readily accept new foods during this period [47].

### 4.2. Parental Feeding Practices

Throughout childhood, parents are the primary caretakers responsible for shaping their children’s food environment through the foods they choose to purchase and make available, the rules they establish around the timing, frequency, and structure of the meals, and the interactions they have with their children during mealtimes [49]. Parental feeding practices are the strategies used by parents to control or modify what, when, and how much their child eats (e.g., actions such as pressuring the child to eat more, restricting certain foods, or monitoring food consumption) [49]. These practices are influenced by a complex interplay of factors, including socioeconomic characteristics, such as family income and education, cultural background, personality factors, and psychological health [50,51,52,53,54]. Beyond these factors, parents appear to adapt their feeding practices depending on their child’s temperament [55], weight [21,56,57], and eating behaviours [18,19,58,59,60], as well as their perceptions and beliefs about these characteristics [59,61,62,63,64,65,66,67].

### 4.3. How Are These Parental Feeding Practices Assessed and Classified?

For the assessment of the different dimensions of parental feeding practices, many instruments have been developed [68]. The most widely applied tool is the Child Feeding Questionnaire (CFQ) [69], which has been shown to have good psychometric properties in the paediatric population [70]. This questionnaire comprises seven subscales. Four of these measure parental perceptions and concerns about body weight, both their child’s and their own, that may affect parental control of children’s eating: Perceived responsibility (assessing parents’ perceptions of their responsibility for child feeding), Perceived parent weight, Perceived child weight, and Concern about child weight. The other three subscales measure parental feeding practices and attitudes: Restriction (the extent to which parents restrict their child’s access to food); Pressure to eat (tendency to pressure the children to eat more food); and Monitoring (the extent to which parents supervise their child’s eating) [69].

Other popular and validated instruments include The Parental Feeding Style Questionnaire (PFSQ) [71], the Food-Related Parenting Practices Questionnaire [72], the Comprehensive Feeding Practices Questionnaire (CFPQ) [73], the Feeding Practices and Structure Questionnaire (FPSQ) [74], and the Scale of Overt and Covert Control [75]. The latter was conceptualized to distinguish overt and covert restrictive feeding practices. Overt control refers to explicit control over food consumption, such as being firm about what a child should eat, while covert control refers to controlling food intake in a way that cannot be detected by a child, such as avoiding keeping snack foods at home [75].

In 2016, Vaughn and colleagues classified and categorized parental feeding practices documented in the literature into three overarching constructs—coercive control, structure, and autonomy support—with several specific practices within each construct [76]. Coercive control refers to specific control that reflects an attempt to dominate, pressure, or impose the will of the parents over child’s intake. This includes practices such as restricting the child’s access to food, pressuring the child to eat, and using food as a reward or to control negative emotions (instrumental feeding) [76]. Structure is based on the parents’ organization of the children’s eating environment to help them learn and maintain certain dietary behaviours, including parents’ consistent enforcement of rules and boundaries around eating [76]. Structure includes practices such as rules and limits, limited or guided choices, monitoring, role modelling, and food availability and accessibility [76]. Autonomy support is promoting “psychological autonomy and encouragement of independence” [76]. This concerns offering children choices and age-appropriate independent exploration and fostering the child’s capacity to self-regulate when the parent is not present [76]. Autonomy support includes parental strategies such as using logic to persuade children to change their eating behaviours (reasoning); using positive reinforcement through verbal feedback (praise); positively, gently, and supportively encouraging their children to adopt healthy eating habits (encouragement); involving the child in meal planning and preparation (child involvement); providing their children with information and skills to help them make informed food choices (nutrition/food education) [76].

### 4.4. Association with Children’s Eating Behaviours

The association between parental feeding practices and children’s eating behaviours has been extensively investigated (Table 2), with a focus on understanding how parents can influence their children’s eating behaviours. For a better organization, the description of these associations will be carried out, according to the Vaughn and colleagues’ classification and categorization of parental feeding practices [76], as follows.

#### 4.4.1. Coercive Control Practices

Coercive control practices are among the most studied and appear to have more negative than positive effects [76]. It has been hypothesized that these practices may impair children’s ability to self-regulate and respond appropriately to their internal signals of hunger or satiety, making them less able to self-regulate food intake [28,77]. In several cross-sectional studies, controlling feeding practices, such as restriction, were mostly associated with food approach behaviours, such as a tendency to overeat [49], high food responsiveness [60,77,78,79] and enjoyment of food [16]. In longitudinal studies, restrictive feeding practices from 5 to 9 years of age were associated with increased eating in the absence of hunger (measured in a laboratory setting by giving children free access to a variety of palatable snacks after a standard lunch) [28]. At 2 years of age, maternal restriction was associated with a tendency to overeat, alongside other food approach behaviours (such as a more avid appetite and enjoyment of food) one year later [16]. On the other hand, pressure to eat was associated with behaviours such as high food fussiness and low food responsiveness and enjoyment of food in cross-sectional [77,78] and longitudinal studies [80]. Instrumental feeding (the use of food to reward children’s behaviour) was associated with eating in the absence of hunger [81].

#### 4.4.2. Structure

Regarding structure, there is evidence suggesting that this practice may promote healthier children’s diet-related outcomes. Structured, family-meal setting was related to increased levels of self-regulation in eating among preschool-aged children [82]. Mealtime structure was related to low food fussiness [83,84]. Monitoring (the extent to which parents track what and how much the child eats) [69] was negatively associated with the tendency to overeat and other food approach behaviours [77,85], even using a longitudinal study design [16]. Modelling of healthy eating predicted low child food fussiness and high interest in food one year later in a sample of 157 children aged 2- to 4-years-old [80].

#### 4.4.3. Autonomy-Supportive Practices

Autonomy-supportive practices, such as the involvement of the children and encouraging them to try new foods, were associated with healthier food choices [86,87,88]. However, the association with eating behaviours has been less examined. In a study of 111 2-to 4- year-old children, practices such as involving children in food preparation and teaching children about nutrition were related to greater enjoyment of food and less food fussiness [89]. Low levels of food fussiness were associated with great maternal encouragement of balance and variety in children aged 3 to 6 years [90]. In a qualitative study, parents of children (aged 2 to 5 years) with healthy food preferences reported using strategies such as encouraging children to try new foods and involving children in food selection and preparation [91]. A few randomized control trials have examined the effect of interventions targeting parental feeding practices on children’s eating behaviours. In the INSIGHT Responsive Parenting intervention, mothers were taught how to recognize and respond appropriately to hunger and fullness cues and how to adopt structure-based and non-controlling feeding practices (intervention group *n* = 140, control group *n* = 139). The intervention content was delivered in four home visits during the first year after birth [92]. Results have shown that this intervention led to decreased use of controlling feeding practices, such as pressure to eat and instrumental feeding, and an increase in structure-based feeding practices, such as consistent meal routines, at ages 1 and 3 years [92,93]. Regarding eating behaviours, emotional overeating was the only behaviour that differed between groups (lower in children from the intervention group). However, a moderation effect was observed for satiety responsiveness and food responsiveness on some maternal feeding practices (i.e., for children with lower levels of food responsiveness, mothers from the control group used more pressure than mothers from the intervention group) [92]. In the NOURISH randomized controlled trial, the intervention comprised two modules of six sessions, delivered over 12 weeks (intervention group *n* = 352, control group *n* = 346) [94]. Parents were guided on responsive feeding practices and how to promote healthy food intake and limit exposure to energy-dense, nutrient-poor foods to support the development of healthy food preferences [94]. This intervention resulted in small but significant effects on children’s eating behaviours. Children from the intervention group showed lower food responsiveness and higher satiety responsiveness up to 3.5 years post-intervention [95].

### 4.5. Reciprocal Influences between Parental Feeding Practices and Eating Behaviours

Despite the literature to date being mostly focused on the influence of parental feeding practices on children’s behaviours, it is also acknowledged that these behaviours can affect the feeding practices used by parents. Recent studies have examined this hypothesis and tested the direction of influence, whether parents influence children, children influence parents, or if there is a mutual influence [14,15,21,22,23]. These studies have used cross-lagged models to examine the causal influences between variables [96]. Variables can influence each other (bidirectional relationship), or one variable can influence another without mutual influence (unidirectional relationship). A study of 4845 mother–child dyads from the population-based Generation R cohort found bidirectional associations between pressure to eat and food fussiness, from 18 months to 6 years of age [21]. However, the strongest path was found to be from fussy eating at 3 years old to pressure to eat one year later [21]. In the Generation XXI cohort (*n* = 3698), from 4 to 7 years of age, pressuring practices were also bidirectionally associated with food refusal and eating small amounts of food [23]. This suggests that despite parents’ intentions to increase their child’s food consumption by pressuring them to eat more, this practice seems to be counterproductive and worsens existing food avoidance behaviours. Another study, on 187 children aged 4- to 5 years old, examining an 18-month period, found that observed prompts to eat (directing the child to consume a food different from the food that they are eating) were associated with eating in observance of hunger, but no association was found in the opposite direction or for any other forms of pressure [97]. This suggests that different forms of pressuring a child to eat may lead to different results and highlights the importance of a more in-depth examination of this practice. Nevertheless, this also advocates that pressure to eat may be counterproductive [97]. Additionally, a study in infants, evaluating three-time points from 3 to 12 months of age, found that food avoidance was prospectively associated with higher parental persuasive feeding (feeding the infant even if they are not hungry), but this practice did not predict infants’ behaviours [98].

**Table 2 healthcare-11-00400-t002:** Studies on the association between parental feeding practices and children’s eating behaviours.

First Author, Year	Country	Sample	Design	Results
Webber, 2010 [60]	UK	531 children aged 7 to 9 y	Cross-sectional	Restriction was positively associated with FR. Food avoidant behaviours were positively associated with pressure to eat.
P. W. Jansen, 2012 [77]	The Netherlands	4987 children aged 4 y	Cross-sectional	Monitoring was inversely correlated with SR, FF, EOE, FR, and positively correlate with EF. Restriction was positively correlated with EUE, SR, FF, EOE, FR. Pressure to eat was positively correlated with EOE, SR, FF, EOE, and negatively correlated with FR and EF.
Carnell, 2014 [78]	UK	439 children aged 3–5 y	Cross-sectional	Restriction was associated with increased FR. Pressure to eat was associated with higher SR, but not prompting to eat. Both instrumental and emotional feeding were associated with higher FR.
De-Jongh González, 2021 [79]	Canada	565 children aged 5–12 y	Cross-sectional	Restriction was positively associated with EOE and FR. Accommodate the child and child involvement were negatively associated with FF.
Rodgers, 2013 [16]	Australia	323 children aged 1.5–2.5 y	Longitudinal (follow-up 1 y)	Restriction was prospectively correlated with tendency to overeat. Monitoring was negatively associated with emotional eating and food approach behaviours. Encouragement to eat was related to higher food approach. Control and emotional feeding were associated with increased tendency to overeat.
Birch, 2003 [28]	USA	192 girls aged 5–9 y	Longitudinal (follow-up 2 y)	Restriction predicted higher eating in absence of hunger.
Gregory, 2010 [80]	Australia	156 children aged 2–4 y	Longitudinal (follow-up 1 y)	Modelling of healthy eating predicted lower FF and higher interest in food, and pressure to eat predicted lower interest in food.
Frankel, 2018 [82]	USA	379 children mean age 4.1 y	Cross-sectional	Structured meal setting and family meal setting, but not structured meal timing, were associated with children’s heightened levels of self-regulation in eating.
Monnery-Patris, 2019 [81]	France	45 children aged 1–5 y	Cross-sectional	Use of food as reward was associated with increased eating in the absence of hunger. Parental awareness of children’s cues was negatively associated with behaviour and with poor eating compensation ability.
F.Powell, 2017 [83]	UK	75 children aged 2–4 y	Cross-sectional	Mothers who eat with their child and eat the same food have children who refuse fewer foods and are easier to feed. Giving children choice in meals and portion size leads to more positive mealtime experiences.
Finnane, 2017 [84]	Australia	413 children aged 1–10 y	Cross-sectional	Persuasive feeding was associated with higher SR, SE, FF, and EUE, and lower EF. Reward for behaviour was linked to lower SR, and higher FR, EF, and emotional eating. While reward for eating was linked to higher SR, FF, and lower EF. Structured meal setting was linked to higher EF. Family meal setting was related to lower FF and higher EF.
Warkentin, 2020 [85]	UK	70 children aged 3–5 y	Cross-sectional	Higher pressure to eat was associated with slower eating rate, while higher monitoring was associated with decreased eating in absence of hunger.
Holley, 2020 [89]	UK	111 children aged 2–4 y	Cross-sectional	Encouraging balance and variety, providing a healthy home environment, involving children in food choice and preparation, and teaching the child about nutrition was associated with lower levels of FF and higher EF.
F.C. Powell, 2011 [90]	UK	104 children aged 3–6 y	Cross-sectional	Controlling feeding practices, the use of food for behaviour regulation, low encouragement of a balanced and varied food intake, and low provision of a healthy food-related home environment were associated with food avoidant behaviours.
Russell, 2015 [91]	Australia	57 children aged 2–5 y	Cross-sectional	Parents of children with healthy food preferences reported using strategies such as encouraging children to try new foods, exposing the child to foods, and involving children in food selection and preparation.
Ruggiero, 2021 [93] Savage, 2018 [93]	USA	intervention group n = 140, control group n = 139	Randomized clinical trial (follow-up 1, 1.5, 2, 2.5 y)	The intervention group used more consistent meal routines, less pressure, and less use of food as rewards or to soothe compared to the control group. Children from the intervention group had lower EOE at 2.5 years. No other study group differences were found.
Daniels, 2012 [94] Magarey, 2016 [95]	Australia	intervention group n = 352, control group n = 346	Randomized clinical trial (follow-up 6 m, 2 and 3.5 y)	Intervention mothers reported less frequent use of nonresponsive feeding practices. Up to 3.5 years post-intervention, children in the intervention group showed lower FR and higher SR; differences were small.
Steinsbekk, 2016 [15]	Norway	797 children aged 6 y	Longitudinal (follow-up 2 y)	Instrumental feeding at 6 y of age predicted increased EOE and FR 2 years later, whereas greater encouragement to eat predicted increased EF. No evidence of child effects was found.
Jansen, 2017 [21]	The Netherlands	4845 children aged 1.5 y	Longitudinal (follow-up 1.5, 2.5, 4.5 y)	Fussy eating (at 1.5 and 3 y) predicted higher levels of pressure to eat at age 4 y, and pressure to eat at 4 y also predicted more fussiness at age 6 y. In a path analysis, the relationship from fussy eating at 3y to pressure to eat one year later was the strongest.
Berge, 2020 [22]	USA	534 children aged 2–4 y	Longitudinal (follow-up 1, 2, and 3 y)	A bidirectional positive association was observed between instrumental feeding and FR. Greater emotional feeding was associated with greater SR from baseline to year 1, and greater emotional feeding was associated with lower SR from year 1 to year 2. Emotional feeding predicted greater FR.
Costa, 2021 [23]	Portugal	3698 children aged 4 y	Longitudinal (follow-up 3 y)	Eating large amounts of food predicted higher Restriction, all other associations had bidirectional effects. Eating large amounts of food and food refusal predicted parental feeding practices such as Perceived Monitoring and Pressure to eat at age, but these practices were also prospectively linked to these eating behaviours.
E. Jansen, 2018 [14]	Australia	207 children aged 2 y	Longitudinal (follow-up 1.7 and 3 y)	Higher reward for behaviour and lower covert restriction at 2 y were associated with FR at 3.7 y. Meal setting predicted lower SR from 3.7 y to 5 y. SR predicted increased structured meal timing, and overt and covert restriction.
Galindo, 2018 [97]	USA	187 children ages 4–5 y	Longitudinal (follow-up 18 m)	Observed maternal prompts to eat a different food predicted eating in absence of hunger. No association was observed for pressure to eat.
Burnett, 2022 [98]	Australia	380 children aged >6 m	Longitudinal (follow-up 3, 6 m)	Food avoidance at 6 m was associated with higher parental persuasive feeding (feeding the infant even if they are not hungry) at 9 m. However, this practice did not predict either food avoidance or food approach behaviours.
P. W. Jansen, 2020 [99]	The Netherlands	3642 children aged 4 y	Longitudinal (follow-up 5 y)	Use of food as reward predicted EOE and picky eating at age 9 y. On the opposite direction, higher EOE and FR predicted more use of this practice. No association was observed with SR.

y: years old, m: months, EF: Enjoyment of food, FR: Food responsiveness, SR: Satiety responsiveness, EOE: Emotional overeating, SE: Slowness in eating, EUE: Emotional undereating, FF: Food fussiness.

In the study from Generation XXI, practices such as monitoring the child’s food consumption were negatively associated in both directions with eating large amounts of food and food refusal (i.e., parents who used these practices reported less that their children were eating large amounts of food and refusing to eat) [23]. These associations suggest that monitoring strategies may support the establishment of desirable eating behaviours in childhood.

Regarding restriction, in Generation XXI, parents of children who were eating large amounts of food at age 4 restricted the children’s access to foods (by ensuring that the child does not eat these foods and avoiding buying them) and were more concerned about the child’s weight at age 7, but no effect of these practices on children’s behaviour was observed [23]. Jansen et al., in a sample of 207 Australian children, reported an association between covert restriction at age 2 and low food responsiveness from 2 to 3.7 years of age, but not from 3.7 to age 5 [14]. This suggests that the effect of these types of practices may be more noticeable at younger ages. However, it is difficult to directly compare the results since these studies have used different methods to evaluate and analyse parental feeding practices.

In a study of 797 children from a Norwegian cohort, instrumental feeding at age 6 years predicted emotional overeating and food responsiveness at age 8 [15], whereas, in a sample of 479 low-income children, this practice was bi-directionally associated with food responsiveness from age 1 to 3 years (high food responsiveness predicted instrumental feeding and vice versa) [22]. In E. Jansen et al. (2018), high reward for behaviour was prospectively associated with food responsiveness at age 3.7 years, but from age 3.7 to age 5, these associations were no longer significant. In another study, from the Generation R cohort, the use of food as a reward was found to predict emotional overeating and picky eating from ages 4 to 9, but, also, high levels of emotional overeating and food responsiveness were related to more use of food as a reward [99]. Despite some inconsistencies in the findings, these studies suggest that instrumental feeding practices may promote the development of food approach behaviours in childhood. It is plausible that offering certain foods (usually highly palatable) as rewards may increase children’s preference and interest in these foods.

### 4.6. A Model for the Development of Children’s Eating Behaviours: The Role of Parental Feeding Practices

Russel and Russell have developed a model to conceptualize the biological and psychosocial factors that influence children’s eating behaviours [10]. Figure 1 presents a simplified version of the original model more focused on the concepts of the present review. This model focuses on the characteristics and behaviours of parents and children, as well as their interactions and influences over time [100]. Accordingly, biological foundations of appetite are those related to genetic predispositions and other biological bases of the children, such as their temperament [10]. Psychosocial influences involve parental factors such as parental feeding practices and other elements of the family and home environment [10]. In short, parental characteristics, including biological, psychosocial, environmental, and cultural factors, and child characteristics, such as the child’s biological foundations, influence parental cognitions (e.g., beliefs about children and diet), expectations (e.g., about children and their behaviours), and interpretations that influence parents’ behaviours and practices. These parenting behaviours and practices may, in turn, affect children’s outcomes, namely their eating behaviours. Therefore, earlier interactions or behaviours of the parent and child can influence subsequent behaviour in the other, creating a bidirectional relationship [100]. For example, if parents, in response to the perception that a child is eating very little, try to use pressure strategies to get their child to eat more, these practices may lead to worsening food refusal over time. Or, if a child has a predisposition to eat in response to emotions, parents may offer food to calm the child or control their behaviour; however, this practice may foster the development of emotional overeating.

## 5. Conclusions and Future Directions

Parents have a direct impact on their child’s diet through the quantity and quality of food they provide and the feeding practices they adopt. Studies have shown an association between the early feeding environment, parental feeding practices, and children’s eating behaviours. However, the relationship between parents and children is likely reciprocal, as children’s behaviours also influence parental choices [14,15,21,22,23]. The extent to which feeding practices or children’s behaviours can be considered the main drivers of influence depends on the type of practices and the age of the children, which makes it difficult to draw general conclusions. However, coercive practices, such as restriction, instrumental feeding, and pressure to eat, are generally associated with less desirable eating behaviours, while structure and autonomy-supportive practices have been linked to more favourable outcomes.

Limitations in the literature include a lack of longitudinal studies with long follow-up periods and studies examining parental feeding practices in a more naturalistic manner, such as independent observations of feeding practices during mealtimes. Most studies rely on self-reported measurements, which are the most practical method for assessing behaviours in large-scale studies, but have been found to have poor associations with independent observations [94,95]. The same applies to children’s eating behaviours, however, previous studies have shown consistent associations between parental reports and behavioural measures of eating [24,101].

There is some evidence on the effect of parental feeding practices from randomized control trials. Interventions conducted during the first years of life led to small improvements in some children’s eating behaviours [87,90]. However, as children grow and develop, parents face different challenges regarding feeding. Therefore, long-term interventions that target parental feeding practices at different developmental stages should be explored to fully understand the cause-and-effect relationship and evaluate the most effective strategies for promoting a healthy development of children.

As a general message, parents should focus on creating a healthy food environment (consistently and repeatedly offering children healthy foods) and avoiding the use of coercive practices (i.e., attempting to dominate, pressure, or impose their will) in response to their children’s eating behaviours. Parents should also prioritize responsive feeding practices that respect children’s hunger and satiety cues. It is important to highlight that eating behaviours have multiple determinants, including genetics, and certain behaviours (as well as an unhealthy weight status) can be challenging for parents to deal with. Therefore, it is relevant for healthcare professionals to educate parents on the most effective practices to use, taking into consideration both parent and child characteristics and the family dynamics.

## Figures and Tables

**Figure 1 healthcare-11-00400-f001:**
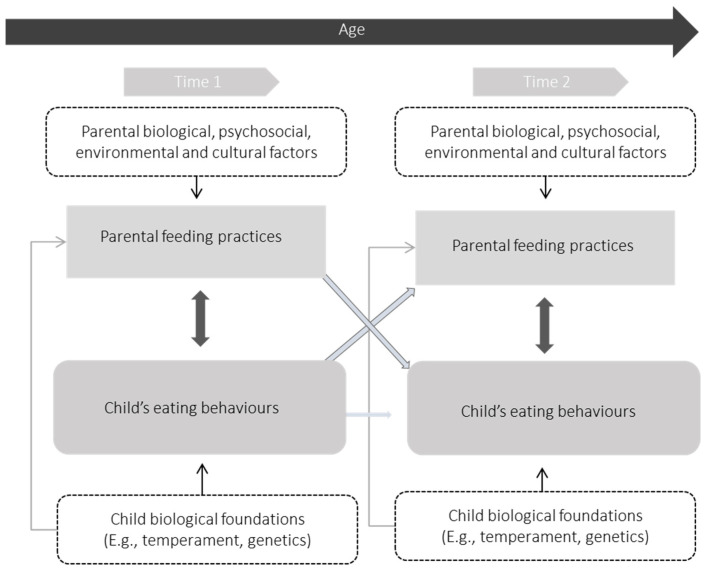
A model of biological and psychosocial processes in the early development of child’s eating behaviours. Adapted from Russell and Russell, 2018 and Russell and Russell, 2019 [10,100].

**Table 1 healthcare-11-00400-t001:** Different methods and measures to assess eating behaviour in children.

Eating Behaviour	Method(s) of Assessment	Description
Eating in the absence of hunger (EAH)	Direct observation in a laboratory setting [27].	Eating when satiated in response to the presence of palatable snack food [28].
Eating rate	Direct observation of (usually) videotapes [29].	Total energy or mouthfuls of food consumed within a given time interval [30].
Compensation of energy intake	Compensation trials [31]	Adjustments in intake in response to changes in the caloric content of a preload (fixed amount of food or nutrient) after a predetermined time delay [32].
Enjoyment of food	CEBQ [26]	General interest in food and the amount of pleasure experienced when eating [33].
Desire to drink	CEBQ [26]	Desire to drink liquids, particularly sugar-sweetened beverages [33].
Food responsiveness	CEBQ [26]	Eating in response to food cues (such as the sight or smell of food) [33].
Satiety responsiveness	CEBQ [26]	The ability to recognize and adjust eating in response to internal feelings of satiety or fullness [33].
Emotional overeating	CEBQ [26]	Undereating in response to negative emotional states [33].
Slowness in eating	CEBQ [26]	Eating slowly during a meal [33].
Emotional undereating	CEBQ [26]	Overeating in response to negative emotional states [33].
Food fussiness	CEBQ [26]	Being highly selective of foods [33].

## Data Availability

Not applicable.

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
