# Peer review of "Parental Feeding Practices and Children’s Eating Behaviours: An Overview of Their Complex Relationship"

_healthcare, 2023, doi:10.3390/healthcare11030400_

Round 1

Reviewer 1 Report

Introduction     

In the first paragraph it would be useful to explain the associations between different appetites and dietary intake, i.e. do smaller appetites eat less vegetables?

Further rationale could be given in the opening paragraphs, why are the early feeding environment and parent feeding practices the focus?

Information on the perimeters of the review could be given. What was the decision to include certain eating behaviours and feeding practices? Was it based on theory or frequently examined eating behaviours and feeding practices?

Section 3.1

Line 103 – please rephrase to accept new foods more readily

The introduction seems to jump topic without any leading or connecting sentences between sections, it would be helpful to have some rationale to help guide the reader from early feeding environment to parental feeding practices. 

Section 3.2

Can you briefly give some examples of parental feeding practices.

Lines 117-121 – Move this paragraph to section 3.4, it seems out of place here

Section 3.3

The FPSQ is another widely used instrument to assess parental feeding practices that should be mentioned.

Section 3.4

It would be good to outline what ages these studies focus on, sometimes it is mentioned, however other times it is not (i.e., Lines 181-190).

Line 187-188, this seems to stray from the aims (feeding practices and child eating behaviours), as this sentence is discussing the associations with

Section 3.5

There are a couple of studies that could be added to this section:

Galindo L, Power TG, Beck AD, Fisher JO, O'Connor TM, Hughes SO. Predicting preschool children's eating in the absence of hunger from maternal pressure to eat: a longitudinal study of low-income, Latina mothers. Appetite. 2018 Jan 1;120:281-6.

Jansen PW, de Barse LM, Jaddoe VW, Verhulst FC, Franco OH, Tiemeier H. Bi-directional associations between child fussy eating and parents' pressure to eat: Who influences whom?. Physiology & behavior. 2017 Jul 1;176:101-6.

Burnett AJ, Jansen E, Appleton J, Rossiter C, Fowler C, Denney-Wilson E, Russell CG. Bidirectional associations between parental feeding practices, infant appetitive traits and infant BMIz: a longitudinal cohort study. International Journal of Behavioral Nutrition and Physical Activity. 2022 Dec;19(1):1-1.

Section 3.6

A model for the development of children’s eating behaviours: the role of parental feeding 274 practices, should this be section 3.6?

Line 277 – Rephrase to Figure 1 presents a simplified……

Conclusion

The self-report nature of children’s eating behaviours is also a limitation.

The conclusion could be stronger, the ‘so what’ of the review needs to strengthen.

Reviewer 2 Report

This narrative review provides discusses parental influences on children’s eating behaviors as well as reciprocal influences.  The information in the review is potentially useful to practitioners and researchers. 

However, it is not clear how authors conducted their literature search. While this is not a systematic review, there should be some description of how the literature was searched, considerations for inclusion and exclusion, dates, etc. It is not clear how thorough this review is. A PRISMA diagram is not necessary, but it would be helpful to know how the search was conducted.

I think there is somewhat of a disconnect between the title and the stated purpose. The primary purpose stated is to review parental feeding practices. However, the title implies the review is about reciprocal associations between children’s eating behaviors and parental feeding practices, which is covered in the review. That said, only 7 studies are described in the section on Reciprocal influences, which is the stated title of the paper. It seems like there would be many more. If not, this can be stated with a description of methods for literature search.

I personally found the readability of the manuscript somewhat difficult. I think this is partially because some sections were very limited in literature described (children’s eating behaviors for example) and the connections between sections were not clear.

The purpose of Section 2 on Children’s eating behaviors is unclear. I suggest expanding it to include discussion of some studies. A summary of literature on children’s eating behaviors that is similar to the summary of parental feeding practices (Section 3.2) might be helpful. What about the studies referenced in Table 1?

In section 3.4, I suggest subheadings for each topic – coercive control, structure, autonomy. I would also suggest indicating that you are using the constructs suggested by Vaughn et al as the basis for discussion. 

Additionally, tables would be helpful summarizing the literature on each section or each topic with type of study, population included, variables, results, etc.  For instance, Table 1 could be expanded.

Line 323-324: From this review, it’s not clear what you mean by "parents not only focusing on children’s characteristics." And to be honest, I am not clear what children’s characteristics are being referred to. Also, state that current evidence suggests, not given current evidence.

Round 2

Reviewer 2 Report

Thank you for your responses. I appreciate your hard work in providing revisions. One suggestion would be to organize the table in some logical way, such as listing all similar types of studies together (cross-sectional, longitudinal etc) and then either in alphabetical order or by reference number.